# Quality of Life of Gynaeoncology Patients and Family Caregivers

**DOI:** 10.3390/ijerph19116450

**Published:** 2022-05-26

**Authors:** Raimi Zamriah Hasan, Beng Kwang Ng, Su Ee Phon, Abdul Kadir Abdul Karim, Pei Shan Lim, Abdul Ghani Nur Azurah

**Affiliations:** 1Department of Obstetrics and Gynaecology, Hospital Melaka, Jalan Mufti Haji Khalil, Melaka 75400, Malaysia; raimi.z84@gmail.com; 2Department of Obstetrics and Gynaecology, Faculty of Medicine, Universiti Kebangsaan Malaysia, Jalan Yaacob Latiff, Bandar Tun Razak, Cheras, Kuala Lumpur 56000, Malaysia; sephon88@yahoo.com (S.E.P.); abdulkadirabdulkarim@yahoo.com (A.K.A.K.); peishan9900@yahoo.com (P.S.L.); azurah@ppukm.ukm.edu.my (A.G.N.A.)

**Keywords:** quality of life, gynaeoncology, caregiver, cancer

## Abstract

The quality of life of both gynaeoncology patients and their family caregivers is affected by disease. This was a cross-sectional study of gynaeoncology patients and their caregivers in a gynaeoncology clinic and ward in a tertiary centre from 1 November 2017 until 30 April 2018. EQ-5D-5L and SF-36 questionnaires were used for the assessment of quality of life. Sociodemographic characteristics and the quality of life of both patients and caregivers were studied. There were 176 patients approached and 7 patients declined to participate in the study, giving the response rate of 95.9%. A total of 169 patients were recruited and consented to this study. Whereas, for SF-36, five domains that were physical functioning, role limitation due to physical health, energy, pain, and general health were statistically significant between both groups (*p* < 0.05). Factors that favoured a high quality of life in patients were an early stage of cancer and the absence of comorbidities. While for caregivers, being single or educated was associated with a better quality of life. In our study, we found that the quality of life of gynaeoncology patients was lower than their caregivers based on EQ-5D-5L and SF-36 questionnaires.

## 1. Introduction

Approximately 48,635 Malaysian individuals were diagnosed with new cancer in the year of 2020 in Malaysia [1]. Among those, 25,587 were female patients, and ovarian cancer, endometrial cancer, and cervical cancer remained the top five cancers after breast and colorectal cancer [1]. There were 13,929 cancer deaths and 8.5% of them died before the age of 75 years [1]. Compared to other countries as showed in OECD data published in 2019, cancer death in most of the countries ranged from 118 per 100,000 persons in Mexico, 164 per 100,000 persons in Japan, 167 per 100,000 persons in Switzerland, 180 per 100,000 persons in Australia, and to the highest number of 264 per 100,000 persons in Hungary [2]. Regarding ovarian cancer for example, the incidence and mortality rates are expected to globally rise by year 2035. A study published recently by Peremiquel-Trillas et al. showed that the incidence of ovarian cancer might reduce in Catalonia, but the mortality remains higher among older women [3]. Cancer survivors have been known to have a chronic illness trajectory. The disease does not only affect the individuals diagnosed with it, but it also has a major impact to the family members and relative especially the caregivers in various aspects.

Caregiving is a thoughtful and meaningful experience. Previous studies demonstrated some benefits while providing the care, which include post-traumatic growth, an improved sense of self-worth, and increased personal satisfaction [2,3,4,5]. However, most of the studies in recent years reported negative experiences by family caregivers of cancer survivors. The responsibilities of caregiving of cancer patients are multidimensional. These include treatment monitoring, treatment-related symptom management, emotional, financial, spiritual support, and assistance with personal and instrumental care [5]. Caregiving stress will induce numerous predicaments including the restriction of routine activities, psychological stress, marital and family relationships disharmony, and deteriorating physical health [5]. Vanderwerker et al. reported that 13% of caregivers of patients with advanced cancer met the criteria for a psychiatric disorder [6]. Some studies have even reported that levels of depressive symptoms similar to or even greater than the patient themselves [7,8].

Stenberg et al. reported in her literature review that there are more than 200 problems and difficulties, encountered by family caregivers [9]. These will result in a diminished quality of life of the caregivers. The distress suffered by the caregivers will persist over time and may be aggravated by the declining condition of the patient. Both positive and negative experiences of caregivers will subsequently reflect the care provided to the cancer survivors. A detailed assessment needs to be performed in order to optimize the quality of life of the caregivers and this will consequently improve the care provided to the cancer survivors. Clinicians could address the needs of caregivers and nevertheless include this as part of the holistic approach of the patient.

For gynaeoncology patients, their definition and perception of quality of life might differ from the physician and caregivers. Speca et al. demonstrated that patients identified a different set of priority concerns, which were the sense of control over the body, normalcy in everyday life, and the invasiveness of medical interventions [10]. Their quality of life is related to the symptoms and distress experienced by them. The assessment of a patients’ quality of life serves to aid in therapeutic decision making and guiding a treatment plan.

Increasingly, all related professionals and policy makers are keen to utilize the cancer survival information as one of the references in managing various aspects of the cancer treatment and cancer control programmes in the country. To the best of our knowledge, this was the first study conducted in Malaysia to assess the QOL of both gynaeoncology patients and their family caregivers. Thus, our objective of the study was to assess quality of life of both gynaeoncology patients and their family caregivers in a tertiary centre in Malaysia and to identify potential sociodemographic associations of quality-of-life impairments among the caregivers.

## 2. Materials and Methods

### 2.1. Study Design

This was a cross sectional study conducted at a gynaeoncology clinic and gynaeoncology ward in a tertiary centre Malaysia from 1 November 2017 until 30 April 2018. Approval was obtained from the University Research Ethics Committee (Research code FF2017-503) prior to commencing this study. The study population comprised of all gynaeoncology patients and family caregivers. Caregivers in this study includes all direct family members or relatives who directly take care of the affair and daily living of a patient at home. All patients and their caregivers were approached at our gyneoncology clinic and gynaeoncology ward. They were then explained regarding the study and gave a written consent if they agreed to participate in the study. The inclusion criteria were all gynaeoncology patients diagnosed to have gynaecological cancers and all patients except a small group had completed treatment, and family caregivers, who were 18 years old and above, able to understand and read in English, and gave consents to participate in this study. The exclusion criteria were unstable patients, patients with psychiatric problems, cognitive impairments, domestic helpers who act as caregivers, and family caregivers with psychiatric problems or cognitive impairments. There were given two sets of questionnaires, i.e., ED-5D-5L and SF-36. They were then given enough time to fill out the questionnaire and return it to the investigator.

### 2.2. Instruments and Data Collection

EQ-5D-5L questionnaire English version 11 and SF-36 questionnaire English version 12 were used. Both patient and family caregivers answered the questionnaires themselves. Permission was obtained from the authors for its use in this study. The EQ-5D-5L comprised of two pages: the EQ-5D descriptive system and the EQ visual analogue scale (EQ VAS). The descriptive system consisted of five dimensions, which are mobility, self-care, usual activities, pain or discomfort, and anxiety or depression. Each dimension is evaluated by five levels that were no problem, slight problems, moderate problems, severe problems, and extreme problems. This decision results in a 1-digit number that expresses the level selected for that dimension. The digits for the five dimensions can be combined into a 5-digit number that describes the patient’s health state [11]. Level 1 indicates no problem, Level 2 indicates slight problem, Level 3 indicates moderate problem, Level 4 indicates severe problem and, lastly Level 5 indicates unable to/extreme problem. Thus, the higher the score or level perceived indicating poorer quality of life in terms of all these 5 domains, i.e., mobility, self-care, usual activities, pain/discomfort, and anxiety/depression. The EQ VAS can be adopted as a quantitative measure of health outcome that reflects the patient’s own judgement. The EQ VAS records the patient’s self-rated health on a vertical visual analogue scale, which ranges from 0 to 100. One hundred means the best health and 0 means the worst health that the participants could imagine [11].

The Short Form-36 (SF-36) is a 36-item-questionnaire that measures quality of life based on 8 domains. The 8 domains involved are physical functioning, role limitations due to physical health, role limitations due to emotional problems, energy or fatigue, emotional well-being, social functioning, pain, and general health. The scoring is a 2-step process, which involves recoding the scoring key in 2 tables. First table is used to change the original value to score of 0 to 100. High score defines more favourable health state and vice versa. In the second table, items in the same scale are averaged together to create the 8 scale scores. Missing data are not taken into account when calculating the scale scores. Hence, scale scores represent the average for all items in the scale that the respondent answered [12].

### 2.3. Data Analysis

SPSS (Statistical Package for Social Science) version 23 was used for data analysis. Participants’ profiles were presented descriptively in terms of frequency and percentage, or mean and standard deviation, or median and interquartile range, depending on type and distribution of data. Descriptive analyses were done using Student’s *t*-test and Pearson’s Chi-square test. Continuous variables were summarized using means and standard deviation, and differences between the means were examined using the *t*-test. Significant level was set at *p* < 0.05.

## 3. Results

A total of 169 patients were recruited and consented to this study. The demographic data of the patients were shown in Table 1. The median age of patients for this study was 63.0 (55.0, 69.0). The majority of the patients were Malay (56.2%) followed by Chinese (35.5%), Indian (5.3%), and others (3.0%). Most patients were married (87.0%) and more than half of them (67.5%) had other comorbidities, predominantly hypertension (56.8%) followed by diabetes mellitus (36.7%). The majority of the patients (88.7%) had primary and secondary education and 69.2% of patients had an income level less than MYR 2000 (GBP 430).

The majority of the patients (43.2%) were diagnosed with endometrial cancer, followed by ovarian cancer (32.0%), cervical cancer (20.1%), vulval cancer (3.0%), and others (1.8%). Further analysis of the stages of the cancer showed that most of the patients (39.6%) suffered from stage 1 cancer as shown in Table 2. The majority of the patients were diagnosed with malignancy between 1 to 5 years ago (67.5%). Treatment phases varied among these patients with two major groups being post-surgery (34.9%) and post-radiotherapy (35.5%). The other treatment groups were post-chemotherapy (28.4%), palliative (0.6), and prior to surgery (0.6%).

The demographic data of caregivers were shown in Table 3. The median age of the caregivers was 40.0 (33.0, 55.5). The majority of caregivers were Malay (58.6%), followed by Chinese (35.5%), Indian (5.3%), and others (0.6%). Family caregivers were dominated by male caregivers (54.4%) and most of them were spouses (64.1%). Most of the caregivers were married (78.7%) with a secondary level of education (42.6%) and salary range between MYR 2000 to MYR 5000 (63.9%). Almost 90% of the caregivers have been taking care of patients for more than 6 months in duration.

The quality-of-life outcomes assessment based on the EQ-5D-5L questionnaires for both patients and caregivers were demonstrated in Table 4 and Table 5. There were five aspects evaluated, which include mobility, self-care, usual activities, pain or discomfort, and anxiety or depression. The majority of patients and caregivers had no problems in all five aspects. Overall, caregivers have less issue in every five aspects assessed as compared to patients.

Mean VAS scores are shown in Table 6. Patients’ scores were lower compared to their caregivers with a *p*-value of less than 0.001.

The total scores for each variable of SF-36 for both patients and caregivers were shown in Table 7. In general, all domains’ scores of patients were lower than the caregivers. However, five domains that were physical functioning, role limitation due to physical health, energy, pain, and general health were statistically significant between both groups with *p*-value less than 0.05.

The multiple Linear Regression test was used to assess the association between patient’s socio-demographic and clinical factors with the quality-of-life domain scores. The results were shown in Table 8 and Table 9. We found that the significant factors were age, the stage of cancer, and the presence of comorbidities. Higher quality of life scores were associated with an early stage of cancer and the absence of other illnesses. Younger or educated patients had lower quality of life scores in the anxiety/depression domain.

For the caregivers, significant factors associated with quality-of-life domain scores were age, gender, education, income, and the duration of caregiving. These were shown in Table 10 and Table 11. Caregivers, who were male, younger, and had lower income level, had a lower quality of life score. The duration of caregiving was also inversely proportional to the anxiety/depression score. Being educated correlated with a higher score in the general health domain.

## 4. Discussion

The World Health Organization defined quality of life (QOL) as involving a person’s physical health, psychological state, degree of independence, social relationship, personal beliefs, and environment [13]. Quality of life issues are of a great concern and have gained interest in the past decades because effective modern methods of treatment and early diagnosis has led to an increased number of long-term survivors. According to a Malaysian Study on Cancer Survival (MySCan) report published by the Ministry of Health, Malaysia 2018, cervical and ovarian cancer were among the Top 10 cancers diagnosed in Malaysia. The 5-year cancer survival for cervical, ovarian, and endometrial cancer were 46.9%, 51.2%, and 65.1%, respectively [14].

The issues of concern include the impact of cancer on QOL of both patient and family caregivers. Female cancer survivors suffered from psychological, physical, social distress, and a reduced physical well-being, i.e., fatigue, memory loss, a decreased energy level, pain, financial difficulties, and an overall reduced QOL [15,16]. Recent studies showed that newly diagnosed cancer, a low level of education, and being single predict poor QOL [17]. Another study evaluated 137 ovarian cancer survivors demonstrated that the impact of change in QOL scores (i.e., an improvement in appetite, constipation, and global health scores) was significantly associated with an improvement in prognosis of ovarian cancer [18].

In the case of a family caregiver, the burden of caring for their relatives is associated with a significantly reduced physical and psychological health [17]. The medical condition of cancer survivors, associated symptom burden, and physical tasks that the caregivers are expected to perform will affect the physical well-being of the caregivers. As time goes by, caregivers will report worsening physical well-being as the physical demands of caregiving take their toll and the needs of the ill person take centre stage [19]. When a family member is diagnosed to suffer from cancer, there will be inevitably role changes, which are characterized by changes in family members’ expectations and responsibilities. Some might even take leave or vacation from workplace. This will disrupt their financial status adding to the caregiving burden. Some caregivers, who continue their job commitment, would experience difficulty in focusing on their professional tasks.

A prospective quantitative with pre- and post-test design among 30 family care givers showed that providing basic skills training for caregivers led to a significant change in the QOL of the cancer patient [20]. Another meta-analysis and systematic review showed that intervention given to family caregivers of cancer patients in the form of psychoeducation, skills training focused on coping and problem solving, and therapeutic counselling, were essential [21].

In this study, we found that the QOL outcomes (ED-5D-5L) were low in both cancer patients and their caregivers. All the QOL domains (mobility, self-care, usual activities, pain/discomfort, and anxiety/depression) were significantly lower in cancer survivors as compared to their caregivers. This finding was consistent with the study by Awadalla et al. [17]. These might be due to several factors. More than half of our cancer survivors suffer from other chronic diseases, which might further depreciate their quality of life. In terms of salary, the majority of patients have an income less than MYR 2000 (GBP 430), which is regarded as a below average income. Staying in a cosmopolitan city, Kuala Lumpur, with an escalating cost of living is a potential issue that could reduce their quality of life. A study by Nayak et al. in 2017 also demonstrated that financial constraint has a bigger impact of quality of life of both cancer survivors and their caregivers [22].

Based on SF-36 questionnaires, cancer survivors’ scores were lower than caregivers in all the eight domains of SF-36. The lowest score was observed in the domain of general health and role limitation due to physical health. The problematic issues for patients include fear of recurrence, sexual dysfunction due to side effects of treatment such as chemotherapy, body image, fears over potential unexpected pregnancy, and maintaining a household and personal career [23,24,25].

In this study, we found that the age, stage of cancer, and presence of other ailments were strongly related with patient’s quality of life scores. Younger patients were found to be more anxious and depressed. This is consistent with a study done by Chan et al. on the quality of life after gynaecologic cancer treatment [26]. Younger patients are likely to be in denial accepting a diagnosis of cancer and experience more mental health consequences compared with old patients.

Male caregivers had more impaired quality of life scores compared to their female peers. This was in contrast with a study performed by Matthews et al. examining the role and gender differences in cancer-related distress. The study reported a lower quality of life in female caregivers due to their traditional gender role [27]. In Malaysia and other Asian countries, the responsibility of the economic provider traditionally falls on men. With the caregiving burden, men may experience additional role strain particularly in those with a lower socioeconomic status [28].

Pertaining to QOL, our literature review showed that researchers seem to have paid scant attention to the QOL of family caregivers. Although this study was among the earlier study conducted to assess the QOL for both gynaecologic cancer survivors and family caregivers, there were several limitations. First, this was a cross sectional study from a single centre, with a relatively small sample size, the patients were selected because they had family support and we did not record the degree of recovery from cancer. Our findings cannot be generalised for this population of patients in the country. Secondly, health-related quality of life measurements in cancer patients are usually assessed using cancer-specific instruments that are likely to be more responsive than generic instruments [29]. However, in this study, a disease-specific instrument would not allow us to make a comparison between two different subjects, such as cancer survivors and their caregivers. Unfortunately, we did not specify each treatment phase that the patients were having. This should be done as it might affect patients’ and family caregivers’ quality of lives. This is another limitation of this study. Recognize the inherent limitations of the cross-sectional study design specifically with regard to cause–effect relationships. Lastly, we did not assess the potential health risk of the family caregiver due to the stress of caregiving responsibilities.

## 5. Conclusions

Despite the limitations, this study represents an attempt to understand the complicated interaction between cancer survivors and their family caregivers, in terms of their quality of life. In our study, we found that the quality of life of gynaeoncology patients was lower than their caregivers based on EQ-5D-5L and SF-36 questionnaires. Factors that favoured a high quality of life in patients were an early stage of cancer and the absence of comorbidities. While for caregivers, being single or educated was associated with a better quality of life.

## Figures and Tables

**Table 1 ijerph-19-06450-t001:** Socio-demographic data of patient.

	*n* = 169
Age, years	63.0 (55.0, 69.0)
Ethnicity, *n* (%)	
Malay Chinese Indians Others	95 (56.2) 60 (35.5) 9 (5.3) 5 (3.0)
Married, *n* (%)	147 (87.0)
Employed, *n* (%)	80 (47.3)
Type of comorbidities, *n* (%)	
Diabetes Mellitus Hypertension Dyslipidaemia Ischaemic heart disease Stroke Others	62 (36.7) 96 (56.8) 18 (10.7) 3 (1.8) 2 (1.2) 9 (5.3)
Education Level, *n* (%)	
Primary Secondary College Tertiary	57 (33.7) 93 (55.0) 9 (5.3) 10 (5.9)
Income, *n* (%)	
Less than RM2000 (₤430) RM2000 (₤430)–RM5000 (₤1070) More than RM5000 (₤1070)	117 (69.2) 46 (27.2) 6 (3.6)

Data were expressed in median (Quartile) unless specified.

**Table 2 ijerph-19-06450-t002:** Patient’s clinical profile.

	*n* = 169
Type of cancer, *n* (%)	
Ca ovary Ca endometrium Ca cervix Ca vulva Others (vaginal, choriocarcinoma, fallopian tubes)	54 (32.0) 73 (43.2) 34 (20.1) 5 (3.0) 3 (1.8)
Stage of disease, *n* (%)	
I II III IV	67 (39.6) 40 (23.7) 45 (26.6) 17 (10.1)
Duration of cancer, *n* (%)	
Less than 1 year 1–5 years More than 5 years	13 (7.7) 114 (67.5) 42 (24.8)
Current treatment, *n* (%)	
Prior to surgery Post-surgery Chemotherapy Radiotherapy Palliative care	1 (0.6) 59 (34.9) 48 (28.4) 60 (35.5) 1 (0.6)

All data were expressed in Median (Quartile) unless specified.

**Table 3 ijerph-19-06450-t003:** Socio-demographic data of family caregiver.

	*n* = 169
Age, years *	40.0 (33.0, 55.5)
Male Gender, *n* (%)	92 (54.4)
Spouse Son Father	59 (64.1) 31 (33.7) 2 (2.2)
Female Gender, *n* (%)	77 (45.6)
Daughter Mother Sister Granddaughter	64 (83.1) 5 (6.5) 5 (6.5) 3 (3.9)
Ethnicity, *n* (%)	
Malay Chinese Indians Others	99 (58.6) 60 (35.5) 9 (5.3) 1 (0.6)
Married, *n* (%)	133 (78.7)
Education Level, *n* (%)	
Primary Secondary College Tertiary	14 (8.3) 72 (42.6) 57 (33.7) 26 (15.4)
Duration of caregiving more than 6 months, *n* (%)	151 (89.3)
Income level, *n* (%)	
Less than MYR2000 (GBP430) MYR2000 (GBP430)–RM5000 (GBP1070) More than MYR5000 (GBP1070)	47 (27.8) 108 (63.9) 14 (8.3)

* All data were expressed in Median (Quartile) unless specified.

**Table 4 ijerph-19-06450-t004:** Quality of life (QOL) outcomes (EQ-5D-5L) for patient.

QOL	No Problem (*n* in %)	Slight Problem (*n* in %)	Moderate Problem (*n* in %)	Severe Problem (*n* in %)	Extreme Problem (*n* in %)
Mobility	108 (63.9%)	49 (29.0%)	11 (6.5%)	1 (0.6%)	0 (0%)
Self-care	159 (94.1%)	7 (4.1%)	3 (1.8%)	0 (0%)	0 (0%)
Usual activities	134 (79.3%)	28 (16.6%)	6 (3.6%)	1 (0.6%)	0 (0%)
Pain/ Discomfort	101 (59.8%)	53 (31.4%)	13 (7.7%)	2 (1.2%)	0 (0%)
Anxiety/ Depression	121 (71.6%)	33 (19.5%)	12 (7.1%)	3 (1.8%)	0 (0%)

**Table 5 ijerph-19-06450-t005:** Quality of life (QOL) outcomes (EQ-5D-5L) for caregiver.

QOL	No Problem (*n* in %)	Slight Problem (*n* in %)	Moderate Problem (*n* in %)	Severe Problem (*n* in %)	Extreme Problem (*n* in %)
Mobility	159 (94.1%)	10 (5.9%)	0 (0%)	0 (0%)	0 (0%)
Self-care	166 (98.2%)	3 (1.8%)	0 (0%)	0 (0%)	0 (0%)
Usual activities	151 (89.3%)	16 (9.5%)	2 (1.2%)	0 (0%)	0 (0%)
Pain/ Discomfort	138 (81.7%)	30 (17.8%)	1 (0.6%)	0 (0%)	0 (0%)
Anxiety/ Depression	138 (81.7%)	25 (14.8%)	4 (2.4%)	2 (1.2%)	0 (0%)

**Table 6 ijerph-19-06450-t006:** VAS scores for patient and caregiver.

	Patient	Caregivers	*p*-Value
Mean VAS scores	77.22	84.99	<0.001

**Table 7 ijerph-19-06450-t007:** Comparison of 8 domains of SF-36 between patients and caregivers.

SF-36 Variable	Total	Patient	Caregiver	*p*
Physical functioning	81.17 ± 23.9	70.89 ± 25.5	91.45 ± 14.9	<0.001
Role limitation due to physical health	75.52 ± 38.5	63.61± 43.8	87.43 ± 28.9	<0.001
Role limitation due to emotional problem	84.32 ± 34.6	82.84 ± 35.7	85.80 ± 31.0	0.056
Energy	66.07 ± 13.5	64.23 ± 13.3	67.90 ± 10.5	<0.001
Emotion well-being	75.57 ± 11.4	75.01 ± 13.0	76.12 ± 11.2	0.062
Social functioning	87.09 ± 18.8	86.17 ± 20.8	88.02 ± 17.4	0.267
Pain	81.43 ± 20.7	74.08 ± 20.3	88.77 ± 15.6	<0.001
General health	64.20 ± 17.3	59.90 ± 16.6	68.49 ± 15.2	<0.001

**Table 8 ijerph-19-06450-t008:** Association of patient’s socio-demographic and clinical factors with QOL Domain based on EQ-5D-5L questionnaires.

Patient’s Factors	QOL Domain	*b* (95% CI)	*p*-Value
Age	Mobility	0.023 (0.015, 0.031)	<0.001
	Self-care	0.005 (0.001, 0.009)	0.024
	Usual activities	0.001 (0.003, 0.017)	0.008
	Anxiety/depression	−0.016 (−0.025, −0.007)	<0.001
Education	Anxiety/depression	0.137 (0.001, 0.273)	0.048
Income	Usual activities	−0.175 (−0.325, −0.024	0.023
Stage of cancer	Usual activities	0.15 (0.073, 0.228)	<0.001
	Pain/discomfort	0.151 (0.051, 0.250)	0.03
	Anxiety/depression	0.147 (0.046, 0.248)	0.005
Comorbidities	Mobility	−0.326 (−0.529, −0.122)	0.02
	Usual activities	−0.242 (−0.416, −0.069)	0.006
	Anxiety/depression	0.257 (0.033, 0.481)	0.025

*b* = crude regression coefficient.

**Table 9 ijerph-19-06450-t009:** Association of patient’s sociodemographic and clinical factors with QOL Domain based on SF-36 questionnaires.

Patient’s Factors	QOL Domain	*b* (95% CI)	*p*-Value
Age	Physical functioning	−0.775 (−1.095, −0.456)	<0.001
	Physical health	−0.638 (−1.216, −0.591)	0.031
	Energy	−0.253 (−0.426, −0.080)	0.041
	Emotion well-being	0.02 (0.012, 0.896)	0.018
Comorbidities	Physical functioning	14.44 (5.66, 23.12)	<0.001
	Physical health	19.81 (4.57, 35.22)	0.012
	Energy	5.83 (1.23, 10.32)	0.013
	Pain	12.64 (5.71, 19.56)	<0.001
	General health	6.93 (1.12, 12.75)	0.020
Stage	Social	−4.07 (−7.57, −0.66)	0.023

*b* = crude regression coefficient.

**Table 10 ijerph-19-06450-t010:** Association of caregiver’s sociodemographic factors with QOL Domain based on EQ-5D-5L questionnaires.

Caregiver’s Factors	QOL Domain	*b* (95% CI)	*p*-Value
Age	Mobility	0.005 (0.001, 0.008)	0.009
	Pain or discomfort	0.009 (0.003, 0.015)	0.002
Gender	Anxiety/depression	−0.228 (−0.402, −0.055)	0.01
Duration of caregiving	Anxiety/depression	−0.397 (−0.668, −0.125)	0.004

*b* = crude regression coefficient.

**Table 11 ijerph-19-06450-t011:** Association of caregiver’s sociodemographic factors with QOL based on SF-36 questionnaires.

Caregiver’s Factors	QOL Domain	*b* (95% CI)	*p*-Value
Age	Physical functioning	−0.509 (−0.707, −0.310)	<0.001
	Physical health	−0.600 (−1.01, −0.19)	0.004
	Pain	−0.31 (−0.53, −0.09)	0.007
	General health	−0.28 (−0.47, −0.08)	0.006
Gender	Energy	−4.84 (−8.17, −1.51)	0.005
Education	General health	5.01 (1.91, 8.11)	0.002

*b* = crude regression coefficient.

## Data Availability

The data presented in this study are available on request from the corresponding author. The data are not publicly available due to ownership belongs to the institution where the study was conducted.

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
