# Peer review of "Quality of Life of Gynaeoncology Patients and Family Caregivers"

_ijerph, 2022, doi:10.3390/ijerph19116450_

Round 1
Reviewer 1 Report
Overall, this is an informative study. The manuscript could use the help of an English speaking editor to improve the grammar.
Regarding study instruments, it would be helpful to describe what the levels of scores indicate on the EQ-5D. For example, are higher numbers indicative of worse quality of life? Is there a cutpoint for clinically meaningful poor QOL? Same for the EQ VAS – what do the scores indicate?
It would be helpful for international readers to understand equivalents of the RM in Euros, Pounds, and Dollars and/or other currencies.
In the discussion, all patients were referred to as survivors of cancer. The background, methods, and results sections did not make this distinction. The inclusion criteria were “all patients with gynecologic cancer”. In addition, while most of the patient population was described as post treatments, one group was described as “prior to surgery”. Please either correct the discussion section or indicate in the methods section that all patients except for a small group had completed treatment.
Regarding Tables 7-10, it is unnecessary to repeat the factors in the left column. It would helpful to merge the cells. Also, the heading of QOL score is inaccurate. The domains, not the scores are listed in those columns. Please correct the title to state “QOL Domain”.
It would have been interesting to know the health issues / co-morbid conditions of the family caregivers. One important finding from this study not discussed is the potential health risk of the family caregiver due to the stress of the caregiving responsibilities.
Again, this was an important study. Addressing the above concerns would strengthen this manuscript substantially.
Author Response
Please see the attachment. Thank you for your valuable input.

Reviewer 2 Report
In the abstract the same sentences concerning the results (differences in quality of life of patients and caregivers and their determinants) are repeated, please correct the abstract.
Data on the number of people diagnosed with cancer is outdated - please provide more recent data in the introduction.
Introduction lacks epidemiological background and reference to other countries - please complete.
Was the survey randomized?
A small research sample.
The testing methodology is relevant, but the sampling should have been explained in more detail.
The fourth table is not very readable - I suggest improving readability and presenting the results in a different way, perhaps breaking the data into two tables: patient and caregiver?
Some of the conclusions are too obvious, such as that the quality of life of patients is worse than that of their caregivers. This is a natural conclusion that we can make without preparing this type of study.
Other conclusions is too general
Author Response

(The authors gave the same response as above.)

Reviewer 3 Report
May 18, 2022
Title: QUALITY OF LIFE OF GYNAEONCOLOGY PATIENTS AND FAMILY CAREGIVERS
General comments:
Thank you for the opportunity to review this timely article on an important and topic on the QoL in both gynaeoncology patients and their family caregivers.
The authors carried out a cross-sectional study in order to assess quality of life of both gynaeoncology patients and their family caregivers in a tertiary centre, Malaysia and to identify potential sociodemographic associations of quality of life impairments among the caregivers.
Although the study is well-written as well as is quite relevant, I have few concerns about the methodology.
Please find below some comments, suggestions in order to strengthen the potential contribution of this topic in any revision the author(s) might undertake.
Minor revision:
METHODS
Was a sample calculation performed for this study? or was it a non-probability convenience sample? Please justify
What definition of Family caregivers are the authors using to present the results of this study? Explain in the method.
Inform for both instruments if there was a pilot test before data collection, time of application of the questionnaires, how it was applied (self-completion or in the interview format?
It is also important to mention the psychometric analysis of the scales used in this study, minimally the reliability of the scales using Cronbach's alpha.
RESULTS
Table 2
Others (type of cancer) – Which types (state these 3 other tumors)
Limitations
Recognize the inherent limitations of the cross-sectional study design specifically with regard to cause-effect relationships.
Author Response
Please seethe attachment. Thank you for your valuable input.

Round 2
Reviewer 2 Report
1. OK
2. still the data is quite old - 2012 -2016. Isn't there more recent data? More recent data is now available for most countrie
3. Reference only to Korea - not enough, please refer to the whole continent, OECD data, WHO
4. Thank you for the information
5. Thank you for the information
6. OK
7. I still think the tables are unreadable
8. No changes
Author Response
Plesse refer to attachment
